# The Need for Standardizing Diagnosis, Treatment and Clinical Care of Cholecystitis and Biliary Colic in Gallbladder Disease

**DOI:** 10.3390/medicina58030388

**Published:** 2022-03-05

**Authors:** Gerard Doherty, Matthew Manktelow, Brendan Skelly, Paddy Gillespie, Anthony J. Bjourson, Steven Watterson

**Affiliations:** 1Centre for Personalised Medicine, Ulster University, C-TRIC Building, Altnagelvin Hospital Campus, Glenshane Road, Derry BT47 6SB, Northern Ireland, UK; doherty-g29@ulster.ac.uk (G.D.); m.manktelow@ulster.ac.uk (M.M.); brendan.skelly@westerntrust.hscni.net (B.S.); aj.bjourson@ulster.ac.uk (A.J.B.); 2Western Health and Social Care Trust, Altnagelvin Hospital, Glenshane Road, Derry BT47 6SB, Northern Ireland, UK; 3Health Economics & Policy Analysis Centre, National University of Ireland Galway, ILAS Building, Upper Newcastle Road, H91 CF50 Galway, Ireland; paddy.gillespie@nuigalway.ie; 4Northern Ireland Centre for Stratified Medicine, Ulster University, C-TRIC Building, Altnagelvin Hospital Campus, Glenshane Road, Derry BT47 6SB, Northern Ireland, UK

**Keywords:** gallbladder disease, gallstones, biliary colic, cholecystitis, clinical care, cholelithiasis

## Abstract

Gallstones affect 20% of the Western population and will grow in clinical significance as obesity and metabolic diseases become more prevalent. Gallbladder removal (cholecystectomy) is a common treatment for diseases caused by gallstones, with 1.2 million surgeries in the US each year, each costing USD 10,000. Gallbladder disease has a significant impact on the logistics and economics of healthcare. We discuss the two most common presentations of gallbladder disease (biliary colic and cholecystitis) and their pathophysiology, risk factors, signs and symptoms. We discuss the factors that affect clinical care, including diagnosis, treatment outcomes, surgical risk factors, quality of life and cost-efficacy. We highlight the importance of standardised guidelines and objective scoring systems in improving quality, consistency and compatibility across healthcare providers and in improving patient outcomes, collaborative opportunities and the cost-effectiveness of treatment. Guidelines and scoring only exist in select areas of the care pathway. Opportunities exist elsewhere in the care pathway.

## 1. Introduction

Cholelithiasis (gallstone formation, presenting symptomatically as biliary colic) is considered a major public health problem in developed countries, and its symptoms and complications can generate major economic and social burden. Gallstones are masses in the gallbladder or biliary tract formed due to high levels of cholesterol or bilirubin in bile. 

In 2008, it was estimated that 14.2 million women and 6.3 million men had gallbladder disease (GD) in the USA and that 1.2 million cholecystectomies (removal of the gallbladder) are performed each year [1]. Cholecystitis is inflammation of the gallbladder, and the most common cause is gallstones [2]. Cholecystectomy is one of the most common surgical procedures undertaken worldwide [3]. Approximately 66,660 cholecystectomies are performed every year in the UK [4], and annual expenditure on cholecystectomies is estimated at GBP 111.6 million [5]. 

In Germany, 190,000 patients with GD undergo surgery each year [6]. Autopsy reports show that in the UK, 12% of men and 24% of women of all ages have gallstones [7]. It has variously been estimated that gallstones have a prevalence of 10–15% in adults in the United States and Europe [8] and that 20% of the Western population are affected [9], with 75% of adult patients being asymptomatic [8]. 

There is great variability worldwide regarding the known prevalence of gallstones, in part because the disease may be asymptomatic. High rates of incidence occur in the United States, Chile, Sweden, Germany and Austria, whereas Asian populations appear to have the lowest incidence of gallstone disease [10,11,12,13]. Incidence approaches 30% in the people of Santiago, Chile, whilst amongst the people of Jiaotong, China, it is 3.5% [14]. 

It is noted that 23.2% of cholecystectomies in Ireland take place in an emergency rather than an elective care setting [15]. Across Europe, this figure is variable, ranging from 9.4% in France to 43.0% in Sweden [16], and this may reflect differences in clinical practice. However, it suggests that improvements may be possible in the quality of care and the efficiency of its delivery if improvements can be made to prognosis and management in a way that facilitates a higher proportion of planned early cholecystectomies. 

## 2. Pathophysiology and Risk Factors 

Acute cholecystitis is the most frequent complication of symptomatic gallstone disease and in 90% of cases is caused by occlusion of the cystic duct, though the neck of the gallbladder may also be occluded [17,18]. Occlusion is usually accompanied by inflammation that in some cases may be bacterial [19], particularly with *H. pylori* [20]. The characteristic sharp continuous pain of symptomatic cholelithiasis is known as biliary colic (BC). 

Gallstone formation is driven by cholesterol supersaturation of bile, nucleation and precipitation of excess cholesterol from biliary micelles and gallbladder hypomotility [21]. Phosphatidylcholine from bile and cholesterol form metastable single lamellar vesicles that are ordinarily converted into stable mixed micelles. However, when the bile acids are supersaturated, lamellar vesicles remain that fuse into large unstable multilamellar vesicles and nucleate cholesterol crystals. As a result, around 80% of gallstones consist mainly of cholesterol [22]. 

Supersaturation occurs either due to hypersecretion of cholesterol [23,24,25] or when bile acid or phospholipid concentration is reduced, and this can be exacerbated by a lithogenic diet [26,27]. Mucin gel acts as a nucleation matrix for cholesterol crystals [28], occurring in a gallbladder with impaired motility where supersaturation has stimulated pathologic changes in the gallbladder epithelium, inducing abnormal secretion of mucin [29,30]. Gallbladder motility defects are associated with the crystallization of biliary cholesterol [31,32].

If an occlusion persists, the concentrated bile can initiate a chemical cholecystitis. When combined with infection in acute bacterial cholecystitis, the resulting swelling and pain can persist or progressively intensify, leading to fever and a palpable abdominal mass developing [19]. 

Over-eating, low levels of physical activity, obesity, a low-fibre diet, prolonged fasting, rapid weight loss, metabolic syndrome and insulin resistance have all been recognised as risk factors for gallstone formation [9]. There is a relative risk ratio of 1.6 for BMI [33], and worldwide, the proportion of adults with a BMI in excess of 25 kg/m² rose between 1980 and 2013 from 29% to 37% in men and from 30% to 38% in women [34], suggesting that the number of people at risk is likely to continue to grow. 

Liver cirrhosis is a risk factor. Almost 30% of cirrhotic patients have GD, and pigment lithogenesis has been demonstrated following chronic haemolysis and changes to liver metabolism [35]. Hepatitis C infection has been shown to be a risk factor both in patients with liver cirrhosis [36] and chronic hepatitis [35]. 

Little is known about the genetics of GD [37,38]. There is considered to be a genetic contribution of 25–30% in gallstone development, including from the cholesterol transporter ABCG5/G8 and LITH genes [11,39,40,41]. Twin studies [42] and ethnic studies show familial clustering [39]. Ratios of 2.5:1, 2:1 and 3:1 for familial occurrences have been shown for Swedish [43,44], Israeli [45] and Indian populations [46], respectively. A Danish study reported 14/25 monozygotic twins had consistent GD, compared with 6/40 same and 0/36 different sex twins [47]. 

## 3. The Burden of Gallbladder Diseases

Acute attacks of biliary colic and cholecystitis are intensely painful and intermittent or chronic abdominal discomfort is common. The resulting anxiety can be substantial, and the dietary restrictions affect social activity [48]. 

Post-cholecystectomy syndrome refers to a change in symptoms after surgery. Whilst acute upper abdominal pain tends to be resolved by cholecystectomy, gastrointestinal symptoms may persist or even start [49]. Though the balance of evidence suggests that cholecystectomy improves the patient quality of life [50], this question is under examination in a multicentre trial of conservative vs. surgical management, C-GALL [51]. Quality of life improvement has been assessed and scored using validated questionnaires showing regional variation [52,53]. A meta-analysis of 51 studies with patient-reported outcome measures (PROMs) for the quality of life for symptomatic gallstones, including for patients undergoing cholecystectomy, where 78% of studies assessed both pre and post operatively, found a lack of consistency in study design and reporting severely hampered analysis [54]. 

In Europe, GD is the most common of the gastro-intestinal disorders for which patients are admitted to hospital, and this hospitalisation imposes a significant financial burden to health-care providers. In the US, GD is the second most expensive digestive disease, costing USD 6.2 billion per year [55], with inpatient treatment costs estimated to typically exceed USD 10,000 [56]. In 2000, GD was the most common inpatient diagnosis in the US (with 262,411 hospitalizations), and in 2004, 1.8 million ambulatory care visits occurred with a GD diagnosis. GD is associated with the highest socioeconomic costs amongst gastro-intestinal disorders [57].

## 4. Symptoms and Diagnosis

GD symptoms range from the mild and non-specific to severe pain and complications that are specific. Diagnosis is typically based on presentation, haematology and imaging.

### 4.1. Physical Presentation

Patients with symptomatic GD typically present with biliary colic (BC). *Colic* is somewhat of a misnomer, as BC is described as a steady pain, located in the epigastrium and/or right upper quadrant and lasting at least 30 min [58]. Differing guidelines exist for BC diagnosis (see Table 1). The Dutch Association of Surgery (DAS) include two additional descriptions of symptoms associated with BC and GD: pain radiating to the back and a positive response to a simple analgesic [59,60]. The German Society for Digestive and Metabolic Diseases’ S3 guidelines support these and add that symptomatic GD is accompanied by nausea and vomiting [6]. The American Academy of Family Physicians (AAFPI) define BC as a steady pain which rapidly increases in intensity and reaches a plateau, can last for 1 to 5 h and sometimes radiates to the right upper back [61].

In 2007, the first global consensus guidelines for acute cholecystitis (AC) diagnosis and grading were published [62,63,64,65], formulated by the TG07 panel of experts, establishing a global standard. These guidelines have since been refined as TG18. Table 2 presents the TG18 diagnostic criteria, and Table 3 presents the TG18 grading criteria [66].

### 4.2. Haematology

The literature is uncertain with regard to inflammatory biomarkers. NICE refer to CRP in confirmation of acute cholecystitis, while the Tokyo Guidelines for diagnosis refer to both CRP and white cell count (WCC) and differentiate grades with WCC. CRP levels above 198.95 mg/L have been shown to be predictive of grade 3 cholecystitis, whereas CRP levels between 198.95 mg/L and 70.65 mg/L have been shown to be predictive of grade 2 cholecystitis. A mean CRP of 17mg/L has been derived from grade 1 cholecystitis patients [70]. The CRP threshold of 70.65 mg/L for grade 2 cholecystitis has been shown to have a sensitivity of 75% and specificity of 95% [70], and histopathological findings have shown that CRP has better sensitivity than WCC count for cholecystitis [71]. However, there is clinical opinion that WCC is preferable for distinguishing between cholecystitis and BC, as CRP levels will be raised in both, and they can be distinguished with other features (see Table 4) [72].

Laboratory tests belong to the guidelines for pre-operative screening of laparoscopic cholecystectomy (LC) [6,59,60,73,74], but arguments have been made for [6] and against [59,60] their use in diagnosis of symptomatic GD. NICE state that there is sufficient evidence to support using liver function tests in the diagnosis of common bile duct stones (choledocholithiasis) [4,5], and University of Michigan Health System (UMHS) state that the evaluation of gallstones should include lab tests [74]. 

### 4.3. Imaging

Imaging and sonography are used for evaluating acute cholecystitis. The lack of availability and the high costs usually prohibit MRI. Instead, ultrasound is usually preferred due to its speed, accuracy, availability, low cost base, high sensitivity and the existing knowledge base [75]. Ultrasound identifies the presence of stones, distention of the gallbladder lumen, gallbladder wall thickening, a positive Murphy’s sign (provoked by the transducer or the sonographer), pericholecystic fluid and a hyperaemic wall when using a colour Doppler modality [68,76]. In the UK, it is recommended that suspected patients should undergo abdominal ultrasonography and blood tests, including a test of liver function parameters [4,5].

The TG18 criteria for diagnosis of acute cholecystitis require the prior exclusion of chronic cholecystitis and recommend the use of MRI where abdominal ultrasound does not provide a definitive diagnosis, as ultrasound does not necessarily distinguish well between gallbladder wall thickening due to chronic and acute inflammation [66]. The use of T2 weighting in MRI supports the enhanced imaging and assessment of the gallbladder wall when contrast agents and T1 weighting are indicative of acute cholecystitis (see Figure 1).

A wide range of sensitivities and specificities have been found for the use of ultrasound in the diagnosis of acute cholecystitis, though meta-analyses have estimated high accuracy with 80–90% sensitivity and specificity [78,79]. On the other hand, an evaluation of the accuracy of the Tokyo diagnostic guidelines (including confirmatory ultrasound) by their authors found sensitivity of 84.9% and specificity of 50% [80] for acute cholecystitis, not dissimilar to the sensitivity of 74% and specificity of 62% obtained for a similar combination of ultrasound, Murphy’s sign and elevated neutrophils [81]. These specificities are substantially lower than the 83% value attributed to ultrasound alone in near contemporaneous meta-analysis [79]. This variation can be attributed to heterogeneity between the different cohorts of patients examined and variation in diagnostic classification as well as in device and operator characteristics [66]. For example, in the studies mentioned, acute cholecystitis had prevalence of 90.1% and 67%, respectively, with 100% of the patients in Hwang, Marsh and Doyle also having chronic cholecystitis, while for the meta-analysis of Kiewiet, a median of 40% of patients had acute cholecystitis. 

Similarly, ultrasonographic Murphy’s sign has widely varying sensitivity and specificity for diagnosis according to the characteristics of the evaluation performed; depending on criteria and the patient cohort, it has been found to be either highly sensitive or specific for diagnosis of acute cholecystitis, but not both [69]. While clearly the appropriate diagnostic technology for patient assessment [82], it is likely that the judgement of Bree in 1995 still stands: “*The large number of false positives, and only moderate improvement in specificity when accompanied by gallstones, makes this sign unreliable in separating acute from chronic cholecystitis*.” [83].

Patients likely to have a gallstone obstructing the bile duct should receive endoscopic retrograde cholangiography (ERCP) which has a sensitivity and specificity both over 90% [84]. At a low or moderate likelihood, endosonographic or magnetic resonance cholangiography (MRCP) is recommended to determine whether ERCP is required [85,86]. If inflammation of the bile duct (cholangitis) is suspected, blood test results for inflammatory markers and cholestasis markers (such as bilirubin, alkaline phosphatase, gamma GT and transaminase) should be considered [84]. 

## 5. Treatment and Outcomes

Clinical guidelines recommend conservative management for asymptomatic cholelithiasis [4,5,9] except for gallstones > 3 cm, polyps > 1 cm or a calcified “porcelain gallbladder” [18]. Laparoscopic surgery is preferred to open surgery because of the lower risk of bile duct injuries and infection [87]. However, laparoscopic surgery can be converted to open surgery when there are complications such as difficult anatomical identification, excessive bleeding and suspected bile duct injury or choledocholithiasis [88]. Multiple guidelines for surgery exist (see Table 5) [84].

Controversy exists over the use of early LC over open surgery because, although appearing equally safe, LC is believed to have a higher bile duct injury rate [91], though randomised trials have been inconclusive [92,93,94,95]. It has been suggested that outpatient laparoscopic cholecystectomy is safe and feasible, with high levels of patient satisfaction [96]. For admitted patients faced with either surgery or observation, surgery is deemed superior in AC for clinical outcome and shows better cost-effectiveness due to fewer gallstone-related complications. In observation groups, there is a higher rate of readmission and surgery [97]. The suggestion of a “Golden 72 h” for same-admission laparoscopic cholecystectomy is controversial due to the challenge of identifying the time from symptom onset [98].

Guidelines state that LC should not be offered to patients beyond 10 days from onset unless symptoms suggest worsening peritonitis or sepsis, warranting an emergency surgical intervention. It is noted that earlier surgery is associated with shorter hospital stays and fewer complications, but for patients with more than 10 days of symptoms, delaying cholecystectomy for 45 days is better than immediate surgery [97]. 

Elsewhere, it is suggested that patients with AC should have an early cholecystectomy during the first admission if the pain is of less than five days duration or electively following conservative management, approximately six weeks after the acute episode [15]. However, the latter can give rise to post-surgery complications, and it was observed that the 30-day readmission rate for patients who underwent same-admission cholecystectomy was 6.5% compared with 15.1% in those who did not (*p* < 0.001). Failure of index admission cholecystectomy increased the risk of readmissions with an odds ratio of 2.27 [99].

It has been suggested that the treatment strategy should depend on cholecystitis severity, the patient’s general status and underlying diseases [90]. Predictive factors such as response to treatment, Charlson comorbidity index (CCI) [100], an acute physiology score from American Society of Anaesthesiologists—Physical status (ASA-PS) scores [101,102], and the administration of anti-microbials must be reviewed in order to determine the patient’s ability to withstand surgery. Otherwise, conservative management should be considered with biliary drainage if gallbladder inflammation cannot be controlled. For TG18-defined Grade I AC, LC should only be performed if the CCI and ASA-PS scores suggest an ability to withstand surgery; otherwise, surgery should be postponed and conservative management adopted. For Grade II, LC should be considered soon after onset if CCI and ASA-PS scores show the ability to withstand surgery and if the patient is within an advanced surgical centre; otherwise, conservative management and biliary drainage should be considered. For Grade III, normalising organ dysfunction should be considered. 

Percutaneous gallbladder drainage (PCGBD) followed by elective LC has been suggested as a treatment option for patients with AC [103], and recipients had a significantly shorter operation duration (*p* = 0.012). Amongst older patients, endoscopic sphincterotomy (ES) prior to cholecystectomy has been associated with a significant and clinically important reduction in recurrent complications when compared with sphincterotomy alone [104], reducing the risk of recurrent choledocholithiasis (odds ratio 0.38, *p* < 0.001), ascending cholangitis (0.28, *p* < 0.001) and gallstone pancreatitis (0.35, *p* < 0.001). 

Percutaneous cholecystostomy improved clinical outcomes in elderly patients with high risk classification due to comorbidities, reducing hospital stays and morbidity (*p* = 0.002 and *p* = 0.013, respectively) [105]. AC patients who underwent LC later in their admission were more likely to receive open procedures and incur longer postoperative and overall hospitalizations [106].

For BC, medication with nonsteroidal anti-inflammatory drugs is recommended [107]. Spasmolytics or nitro-glycerine can be added and, if pain is severe, opioids may be used [107]. For acute BC, differentiation of immediate analgesia or a causal therapy is advised, and in AC with signs of sepsis, cholangitis abscess or perforation, antibiotics are required [84]. 

## 6. Surgical Risk Factors

The assessment and management of GD varies between hospital, surgeon and country [108,109], underscoring the role of guidelines in reducing inconsistency in care delivery [110]. It has been suggested that the volume of cholecystectomy procedures undertaken at a hospital relates to outcomes [111,112], with the implication that low volume hospitals may be able to improve their quality and cost-efficiency of patient care by working in conjunction with high volume hospitals [113]. 

The presence of inflammation, hypertension, diabetes and previous abdominal surgery has been shown to be statistically significant (*p* < 0.001) in the conversion from laparoscopic surgery to open surgery. Statistically significant independent predictive factors for conversion in patients with AC were sex, age, inflammation, fever, total bilirubin and elevated WBC count [114]. A pre-operative scoring system developed to predict difficult LC in the UK and Ireland found that age, ASA classification, sex, diagnosis of CBD stone or cholecystitis, thick-walled gallbladders, CBD dilation, the use of pre-operative ERCP and non-elective operations were independent predictors of difficulty (see Table 6, Figure 2). A risk score based on these factors returned an area under the ROC curve (AUC) of 0.789 (*p* < 0.001) when externally validated, supporting pre-operative scoring as predictive of difficult LC [115,116].

Scoring has been validated on two large datasets [117]. Higher difficulty grades were associated with conversion to open surgery and a 30-day mortality (AUC = 0.903 and 0.822, respectively). Scoring was found to be predictive of operative duration, conversion to open surgery, 30-day complications and 30-day re-intervention (*p* < 0.001). A scoring system for operative duration was also developed that can be used to optimise surgical efficiency, reduce costs and increase patient and staff satisfaction [118]. Predictive factors associated with prolonged surgeries of >90 min were validated on a second cohort of patients, returning an AUC of 0.708, and were identified as ASA, age, previous surgical admissions, BMI, gallbladder wall thickness and CBD diameter.

Post-operative scoring has been developed that incorporates operative findings, such as the appearance of the gallbladder (GB), presence of GB distention, ease of access, potential biliary complications and the time taken to identify the cystic duct and artery, such that surgical procedures can be benchmarked (see Table 7) [119]. 

## 7. Healthcare Delivery

Delays to care are undesirable for patients, providers and funders and represent missed opportunities for early clinical engagement with the patient [120]. Waiting times are taken as pragmatic and transparent metrics of healthcare performance [121,122]. Nonetheless, delays in access to care for gallbladder disease are commonplace. For example, data for the Republic of Ireland showed 34.7% of patients to be on waiting lists longer than 6 months, with 10.5% longer than 12 months [15]. Delays can be driven by non-clinical factors such as access to equipment, staff and theatre [123]. Waiting times for patients also vary according to the referral pathway used, the hospital and consultant [15]. Such delays only increase challenges for patients and healthcare systems, as readmissions while waiting for treatment have been correlated with poorer outcomes [121,124].

On efficiency grounds, there are clear arguments to be made for the cost effectiveness of improved processes of care. For example, elective LC for symptomatic cholelithiasis has been shown to be inexpensive relative to its associated health gains, although health gains are not experienced uniformly across age groups [125]. Notably, emergency presentation and admission for cholecystectomy is less cost effective than early referral across indications [126]. Furthermore, early referral and surgical intervention within the onset of AC, but before the onset of complications, has been shown to be associated with significant cost reductions [127,128,129,130]. Building on such evidence, it has been suggested that improved clinical decision-making and the streamlining of care pathways could lead to efficiency gains for healthcare systems, both in terms of improved healthcare outcomes and cost savings to healthcare budgets. 

## 8. Discussion

Gallbladder diseases share risk factors with other conditions, including obesity, diabetes and metabolic syndrome, which are likely to be comorbidities. Western trends predict an increasing prevalence of metabolic conditions, and the incidence of GD and BC is likely to increase. Risk factors include lifestyle and diet, and twin studies have shown a role for genetic risk factors. At present, the genetic drivers are not well known, and it is clear that much work is to be done if we are to understand the genetic contribution to GD and BC. The explosion in patient data available from ‘omics technologies and electronic health care records, allied to the growth in data analytics tools means that here are many opportunities in this area. 

As a long-established condition, gallbladder disease is well-characterised. Likewise, the pathologies of biliary colic and acute cholecystitis are well-understood. However, our review of diagnostic guidance indicates that, in normal practice, distinguishing between a mild acute cholecystitis (defined according to TG18) and a severe presentation of biliary colic with associated chronic cholecystitis requires a combination of signs, symptoms and diagnostic markers to achieve acceptable sensitivity and specificity [81]. Given that ambiguous presentations might lead to ambiguous diagnoses, a greater focus on metrics to distinguish between these conditions might better support the evaluation of treatment pathways that differ according to diagnosis. Treatment options are predominantly surgical and demand significant clinical resources. Health economic evaluations on the timing of these options have tended to focus on particular conditions within gallbladder disease, but practical implementation of clinical care pathways justified by such evaluations will necessarily require objectivity and consistency in diagnosis if the gains found by the evaluation are to be achieved. Variation and inconsistency in delivery have implications for patient welfare, staff workload, institutional resources and the costs of delivery. 

A small but growing number of studies are trying to create metrics to assess AC and BC healthcare delivery. Foremost has been the development of the Tokyo guidelines, which have two organisational strengths: (i) they have been developed as a consensus of a global community of expertise, and (ii) they continue to be updated as the clinical understanding of AC progresses. The value of guidelines lies in their ability to create a more objective and standardised delivery of care. This is the first step towards improving care amongst underperforming providers and in creating the consistency and compatibility between providers needed for interaction and collaboration. Similarly, metrics and scoring systems create a more objective and standardised assessment of patients and practices for internal audit. This is the first step towards objective quality improvement in care delivery and cost-efficiency, and it can underpin audit and training activities. It is clear that many aspects of GD and BC healthcare would benefit from guideline and metric development undertaken on the same basis as the Tokyo guidelines. Specifically: (1)The consensus-framed, evidence-based approach taken to diagnosis and treatment of acute cholecystitis should be extended to consider the case definition and aetiology of the related pathologies of biliary colic and chronic cholecystitis.(2)This approach should have regard for operative findings and a patient’s clinical history. Furthermore, they should ideally assess the potential for biochemical or genetic markers to stratify patients according to their operative findings.(3)High-quality evidence should be accumulated on the potential of imaging technologies, leveraging the use of MRI and MRCP that is now commonplace in many health systems.

The benefits of such an approach would include:(1)The relevance to practice of any derived classification of GD;(2)Facilitating optimisation of the treatment pathway, undertaken on exclusion of the urgent pathologies of acute cholecystitis and common bile duct obstruction;(3)A better understanding of the relationship between acute and chronic forms of GD and the implications of this for resource allocation.

## 9. Conclusions

The global high prevalence of gallstone disease makes it a significant health concern that is likely to place a growing demand on healthcare providers. Valuable work has been carried out to create metrics and guidelines to standardise healthcare in this area, but their coverage is limited, and the delivery of care remains highly variable. It is clear that much work remains to create consensus and standardisation in the clinical care pathways and to understand their implications for patient outcomes, for quality and cost-efficiency of care and for the healthcare provider. Here we have summarised much of the work in this area and highlighted where the greatest current need lies for standardisation. This need presents opportunities to further develop our understanding of how we can best deliver healthcare for gallstone diseases.

## Figures and Tables

**Figure 1 medicina-58-00388-f001:**
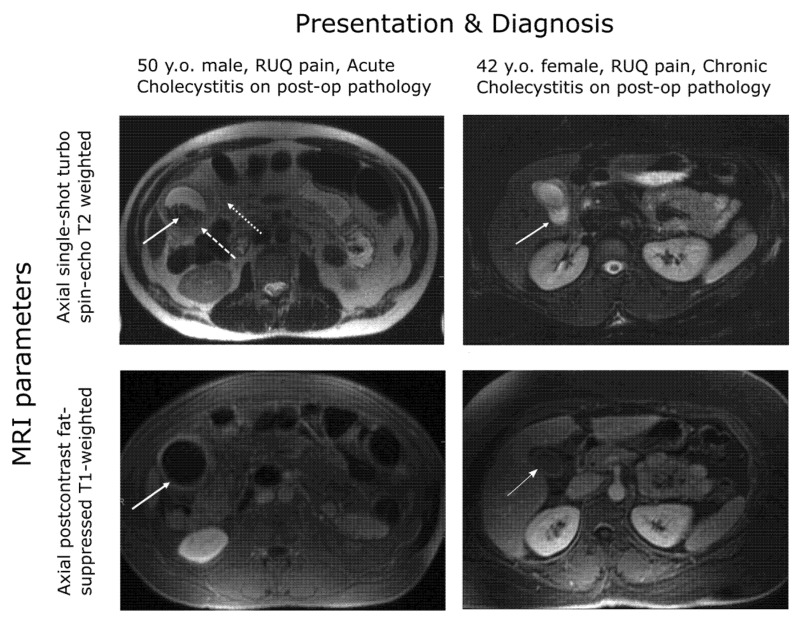
MRI gallbladder imaging. Wall thickening is evident for both the chronic and acute patients but enhanced under contrast only for the acute patient [77].

**Figure 2 medicina-58-00388-f002:**
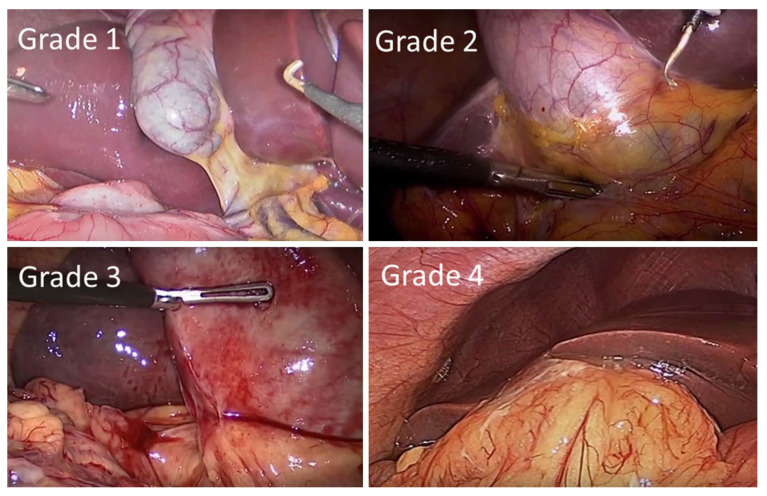
Intra-operative laparoscopic images of the Nassar operative difficulty grades.

**Table 1 medicina-58-00388-t001:** Guidelines describing biliary colic.

Author	Diagnostic Guidelines
Dutch Association of Surgery (DAS) [59]	Pain radiating to back. Positive response to analgesia.
The German Society for Digestive and Metabolic Diseases’ S3 [6]	Biliary colic pain accompanied by nausea and vomiting.
The American Academy of Family Physicians (AAFPI) [61]	Steady pain moderate to severe in epigastrium/right upper quadrant, reaching plateau lasting 1 to 5 h, radiating to upper back at times. If persists with fever and high white blood cell count should raise suspicions of acute cholecystitis, gallstone pancreatitis and ascending cholangitis. Pain in the right upper quadrant of the abdomen; however, pain in this area is not specific for gallstones. The physician must rely on the patient’s description of the pain and on the results of laboratory testing and diagnostic imaging to make a correct diagnosis.

**Table 2 medicina-58-00388-t002:** Tokyo Guidelines for diagnosis of acute cholecystitis, once acute hepatitis, other acute abdominal diseases and chronic cholecystitis have been excluded.

Signs or Symptoms	Conclusion
(Murphy’s Sign *) OR (RUQ ** mass/pain/tenderness)	Local signs of inflammation
(Fever) OR (Elevated CRP) OR (Elevated WCC **)	Systemic signs of inflammation
(Local signs of inflammation) AND (Systemic signs of inflammation)	Suspected diagnosis of acute cholecystitis
(Suspected diagnosis of acute cholecystitis) AND (Imaging findings characteristic of acute cholecystitis)	Definite diagnosis of acute chlecystitis

* Murphy’s sign is a well-known diagnostic indicator for cholecystitis [67,68,69]. The test is performed by asking patients to hold a deep breath whilst the subcostal area of abdomen is palpated. The test is positive if pain occurs on inspiration, denoting inflammation within the gallbladder when it comes into contact with the physician’s hand. ** RUQ: right upper abdominal quadrant, CRP: C-reactive protein, WCC: white blood cell count.

**Table 3 medicina-58-00388-t003:** Tokyo guidelines for grading the severity of acute cholecystitis, TG18.

Severity	Criteria
Grade 1—Mild	Acute cholecystitis not meeting other severity criteria Mild gallbladder inflammation, no organ dysfunction
Grade 2—Moderate	Acute cholecystitis with any of the following but no organ/system dysfunction: Elevated white blood cell count (>18,000/mL) Palpable tender mass at right upper quadrant Duration of complaints exceeding 72 h Marked local inflammation (such as biliary peritonitis, pericholecystic abscess, hepatic abscess, gangrenous cholecystitis, emphysematous cholecystitis)
Grade 3—Severe	Acute cholecystitis with dysfunction of any one of the following organs/systems: Cardiovascular dysfunction (hypotension requiring treatment with dopamine > 5 mg/kg/min of body weight or any dose of norepinephrine) Neurological dysfunction (decreased levels of consciousness) Respiratory dysfunction (ratio of PaO_2_/FiO_2_ < 300) Renal dysfunction (oliguria, creatine > 2.0 mg/dL) Hepatic dysfunction (PT-INR > 1.5)

**Table 4 medicina-58-00388-t004:** Distinguishing features between biliary colic and cholecystitis.

Biliary Colic	Cholecystitis
Spasmodic central epigastric pain, sometimes felt on the right	Constant sharp/stabbing pain in right upper quadrant
No fever, but may have tachycardia if the pain is severe	Pain may radiate to right shoulder and/or back
Tender region over the gallbladder if it is distended	Fever, tachycardia
	Tenderness in the right upper quadrant
	Murphy’s sign—guarding in the right upper quadrant on inspiration

**Table 5 medicina-58-00388-t005:** Differences in recommended treatment programmes.

	Optimal Timing of Treatment after Diagnosis of Acute Cholecystitis	Treatment of Patients with Both Choledocholithiasis and Cholelithiasis	Surgical Strategy
German clinical practice guideline [84]	Laparoscopic cholecystectomy should be carried out within 24 h of hospital admission	Therapeutic splitting (pre- or intraoperatively) is recommended. Cholelithiasis should be treated by cholecystectomy, within 72 h and a stone-free functioning gallbladder can be left in place.	Laparoscopic cholecystectomy using the four-trocar technique both for symptomatic gallstones and in acute cholecystitis
European Association for the Study of the Liver [9]	Cholecystectomy should be carried out preferably within 72 h of admission	Early laparoscopic cholecystectomy should be performed within 72 h of preoperative ERCP.	Laparoscopic cholecystectomy using the four-trocar technique both for symptomatic gallstones and in acute cholecystitis
Society of American Gastrointestinal and Endoscopic Surgeons [89]	Cholecystectomy can be carried out within 72 h of diagnosis	ERCP with stone extraction may be performed either before, during, or after cholecystectomy.	Patients with symptomatic cholelithiasis are suitable for laparoscopic cholecystectomy
Tokyo Guideline 2018 [90]	For both grade I (mild) and grade II (moderate), laparoscopic cholecystectomy should be carried out soon after the onset of symptoms. For Grade III (severe), the degree of organ dysfunction should be determined normalized	N/A	Laparoscopic surgery, even in the presence of severe inflammation (grade III).

**Table 6 medicina-58-00388-t006:** Nassar grading scale for operative findings from the gallbladder, cystic pedicle and associated adhesions.

Grade	Gallbladder	Cystic Pedicle	Adhesions
1	Floppy, non-adherent	Thin and clear	Simple up to the neck/Hartmann’s pouch
2	Mucocele, packed with stones	Fat-laden	Simple up to the body
3	Deep fossa, acute cholecystitis, contracted, fibrosis, Hartman’s adherent to CBD, im-paction	Abnormal anatomy or cystic duct short, dilated or obscured	Dense up to fundus; involving hepatic flexure or duodenum
4	Completely obscured, empyema, gangrene, mass	Impossible to clarify	Dense, fibrosis, wrapping the gallbladder, duodenum or hepatic flexure difficult to separate

**Table 7 medicina-58-00388-t007:** Post-operative grading system for cholecystitis severity.

**Gallbladder Appearance**		Points
	Adhesions < 50% of GB	1
	Adhesions burying GB	3
**Distension/Contraction**		Points
	Distended GB (or contracted shrivelled GB)	1
	Unable to grasp with atraumatic laparoscopic forceps	1
	Stone ≥ 1 cm impacted in Hartman’s pouch	1
**Access**		Points
	BMI > 30	1
	Adhesions from previous surgery limiting access	1
**Severe Sepsis/Complications**		Points
	Bile or pus outside GB	1
**Time to identify cystic artery and duct > 90 min**		Points
	Yes	1
**Total Score vs. Degree of difficulty:-**		**Total score**
	Mild degree of difficulty	<2
	Moderate degree of difficulty	2–4
	Severe degree of difficulty	5–7
	Extreme degree of difficulty	8–10

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
