# Peer review of "The Need for Standardizing Diagnosis, Treatment and Clinical Care of Cholecystitis and Biliary Colic in Gallbladder Disease"

_medicina, 2022, doi:10.3390/medicina58030388_

Round 1

Reviewer 1 Report

The term Biliary Colic is misleading as it is more of a constant pain rather than a colic

This should be better defined 

Author Response

The term Biliary Colic is misleading as it is more of a constant pain rather than a colic. This should be better defined. 

Thank you for your highlighting this ambiguity.  We have altered the section “Physical presentation” to now read,

“Patients with symptomatic GD typically present with biliary colic (BC). “Colic” in this case is somewhat of a misnomer, as it is described as a steady pain, located in the epigastrium and/or right upper quadrant and lasting at least 30min (GREPCO, 1988).  “

Reviewer 2 Report

Some english spelling corrections should be performed

More recent references should be added (ex ESGE and ASGE  guidelines)

Author Response

Some English spelling corrections should be performed.

We apologise for these errors.  We have proofed the manuscript and are confident that they have been removed.

More recent references should be added (ex ESGE and ASGE guidelines)

Thank you for this constructive comment.  Recent references to guidelines on GI endoscopy are now cited in the Imaging section, specifically in the sentence, “At a low or moderate likelihood, endosonographic or magnetic resonance cholangiography (MRCP) is recommended to determine whether ERCP is required (Buxbaum et al., 2019; Manes et al., 2019).” This citations reference the following publications

Buxbaum JL, Fehmi SM, Sultan S, Fishman DS, Qumseya BJ, Cortessis VK, Schilperoort H, Kysh L, Matsuoka L, Yachimski P, Agrawal D (2019), ASGE guideline on the role of endoscopy in the evaluation and management of choledocholithiasis, Gastrointestinal endoscopy 89(6):1075-105.

Manes G, Paspatis G, Aabakken L, Anderloni A, Arvanitakis M, Ah-Soune P, Barthet M, Domagk D, Dumonceau J-M, Gigot J-F, Hritz I, Karamanolis G, Laghi A, Mariani A, Paraskeva K, Pohl J, Ponchon T, Swahn F, ter Steege RWF, Tringali A, Vezakis A, Williams EJ,.van Hooft JE (2019), Endoscopic management of common bile duct stones: European Society of Gastrointestinal Endoscopy (ESGE) guideline, Endoscopy 51(05): 472-491.

Reviewer 3 Report

The present work is a narrative review of the gall bladder diseases, mainly cholycystitis and gall stones.

Recommendations:

Change the title to: "The need for standardizing ....." as the manuscript did not contain any method of standardization.

It is better if the financial part be put under a separate subtitle e.g. "Health economics". In fact, this part should be more highlighted for this manuscript to be more impactful.

Actually, Both health economics + quality of life should be put under one title: "The burden of GB diseases".

No need for 2 or more pages of physiology and pathology. This part should be summarized. The manuscript should focus on the topic in the title. The same applies to the part about risk factors.

Regarding the risk factors; what is the mechanism that HCV infection lead to GB stones?. if this id due to hemolysis; then it is not directly related to HCV. Also the sentence starting from line 194 to line 196 is not correct at all in the present era of DAA drugs. Actually, we are witnessing the marked de-escalation of HCV new cases since the introduction of DAA in the last decade. If this sentence was logic 10 years ago, it is not now. 

Line 169 says: ..... anatomically lean?. Should be "phenotypically" lean.

The link of the clinical trial in line 222 is not working. The trial dated since 2016. Is it published now? because the NIH page says: "this page is no more present".

Also from line 225 to line 227: does this work is "ongoing" since year 1997 and 1999??? Please revise this.!! seems the sentence is copied and pasted from an old reference.!! not proper.!

Regarding comparing different guidelines and recommendations: how can we compare those published in 1998 with those published in 2018. Shouldn't the new guidelines replace the old ones. This needs clarification and or revision. Besides, the tables are not actual comparisons. Some of them are lacking subject of comparison e.g. table (5).

What is the value of table 2 (DD of upper abdominal pain) in the present work??? the manuscript is discussing GB disease not pain in the upper abdomen! Should be removed. It is not the scope of the work here.

Images of what is discussed in the "imaging" section are important to be present. There is no single image in this section.

36 pages and no mention of the definite areas that need clarification or standardization in management of GB stones/cholycystitis (as mentioned in the abstract and also in the conclusion). There should be a final summary (may be in a box) of the main points needing standardization.  

Author Response

The present work is a narrative review of the gall bladder diseases, mainly cholycystitis and gall stones.

We would like to thank reviewer 3 for engaging with the manuscript so deeply and for their highly constructive feedback.  We feel that the recommended changes have significantly improved the manuscript.

Change the title to: "The need for standardizing ....." as the manuscript did not contain any method of standardization.

The title has been amended as suggested.

It is better if the financial part be put under a separate subtitle e.g. "Health economics". In fact, this part should be more highlighted for this manuscript to be more impactful.

Actually, Both health economics + quality of life should be put under one title: "The burden of GB diseases".

We have now created the section, “The burden of gallbladder diseases”.  The paragraph describing costs has been moved to this section along with the paragraphs that were formerly under a “Quality of life” section.

No need for 2 or more pages of physiology and pathology. This part should be summarized. The manuscript should focus on the topic in the title. The same applies to the part about risk factors.

The sections on physiology, pathology and risk factors have been combined into one section, “Pathophysiology and risk factors” and this has been significantly abridged, going from 5 pages in length to 2.

Regarding the risk factors; what is the mechanism that HCV infection lead to GB stones? if this is due to hemolysis; then it is not directly related to HCV. Also the sentence starting from line 194 to line 196 is not correct at all in the present era of DAA drugs. Actually, we are witnessing the marked de-escalation of HCV new cases since the introduction of DAA in the last decade. If this sentence was logic 10 years ago, it is not now. 

The new section on “Pathophysiology and risk factors” has removed much of the discussion around HCV.

Line 169 says: ..... anatomically lean?. Should be "phenotypically" lean.

This line has now been removed as part of creating the new, shorter “Pathophysiology and risk factors” section.

The link of the clinical trial in line 222 is not working. The trial dated since 2016. Is it published now? because the NIH page says: "this page is no more present".

Thank you for identifying this. The text has been corrected to now cite an up to date protocol for this trial published in BMJ Open.  The relevant added reference is:-

 “Ahmed I, Innes K, Brazzelli M, Gillies K, Newlands R, Avenell A, Hernández R, Blazeby J, Croal B, Hudson J, MacLennan G, McCormack K, McDonald A, Murchie P, Ramsay C (2021). Protocol for a randomised controlled trial comparing laparoscopic cholecystectomy with observation/conservative management for preventing recurrent symptoms and complications in adults with uncomplicated symptomatic gallstones (C-Gall trial). BMJ Open. 11(3):e039781. doi: 10.1136/bmjopen-2020-039781.”

Also from line 225 to line 227: does this work is "ongoing" since year 1997 and 1999??? Please revise this.!! seems the sentence is copied and pasted from an old reference.!! not proper.!

We completely agree with this assessment. The text has been deleted.

Regarding comparing different guidelines and recommendations: how can we compare those published in 1998 with those published in 2018. Shouldn't the new guidelines replace the old ones. This needs clarification and or revision. Besides, the tables are not actual comparisons. Some of them are lacking subject of comparison e.g. table (5).

Thank you for such thoughtful and constructive feedback. The GREPCO guidelines have influenced practice since publication and have been superseded. We have removed them from Table 1, though we retain their citation in the body of the text. In old Table 5 (now table 4), our aim was largely to summarise guidelines to support the evaluation of guidelines for differentiation between biliary colic and acute cholecystitis. The cited literature provides commentary on this topic, where formal guidelines focus on the identification of acute cholecystitis with biliary colic as a differential diagnosis.

What is the value of table 2 (DD of upper abdominal pain) in the present work??? The manuscript is discussing GB disease not pain in the upper abdomen! Should be removed. It is not the scope of the work here.

Table 2 was included to indicate the differential diagnoses required to be excluded before the criteria in table 3 is applied, but we concur that it is superfluous, particularly as it does not define the most problematic differential diagnosis, chronic cholecystitis. Table 2 and introductory sentence removed and we believe that this has improved clarity of this section.

Images of what is discussed in the "imaging" section are important to be present. There is no single image in this section.

Thank you for this valuable feedback. Figure 1, composed of four images, has been added, illustrating the utility of MRI for differentiating chronic from acute cholecystitis.  Also added is an explanatory sentence, “Use of T2 weighting in MRI allows assessment of the gallbladder wall, and enhancement of wall imaging when contrast agents and T1 weighting are used is indicative of acute cholecystitis, as shown in Figure 1 below:”

36 pages and no mention of the definite areas that need clarification or standardization in management of GB stones/cholycystitis (as mentioned in the abstract and also in the conclusion). There should be a final summary (may be in a box) of the main points needing standardization.  

Thank you for such constructive feedback. Recommendations and possible benefits have been added to the “Discussion” section and the conclusion section has been amended.